# Reactive Dual Magnetron Sputtering: A Fast Method for Preparing Stoichiometric Microcrystalline ZnWO$_4$ Thin Films

**Yannick Hermans, Faraz Mehmood, Kerstin Lakus-Wollny, Jan P. Hofmann** [ID]**, Thomas Mayer** [ID]
**and Wolfram Jaegermann \***

Surface Science Laboratory, Department of Materials and Earth Sciences, Technical University of Darmstadt, 64287 Darmstadt, Germany; yhermans@surface.tu-darmstadt.de (Y.H.); farazmehmood64@gmail.com (F.M.); kerstin.lakus-wollny@mr.tu-darmstadt.de (K.L.-W.); hofmann@surface.tu-darmstadt.de (J.P.H.); mayerth@surface.tu-darmstadt.de (T.M.)
\* Correspondence: jaegermann@surface.tu-darmstadt.de

**Abstract:** Thin films of ZnWO$_4$, a promising photocatalytic and scintillator material, were deposited for the first time using a reactive dual magnetron sputtering procedure. A ZnO target was operated using an RF signal, and a W target was operated using a DC signal. The power on the ZnO target was changed so that it would match the sputtering rate of the W target operated at 25 W. The effects of the process parameters were characterized using optical spectroscopy, X-ray diffraction, and scanning electron microscopy, including energy dispersive X-ray spectroscopy as well as X-ray photoelectron spectroscopy. It was found that stoichiometric microcrystalline ZnWO$_4$ thin films could be obtained, by operating the ZnO target during the sputtering procedure at a power of 55 W and by post-annealing the resulting thin films for at least 10 h at 600 °C. As FTO coated glass substrates were used, annealing led as well to the incorporation of Na, resulting in n+ doped ZnWO$_4$ thin films.

**Keywords:** ZnWO$_4$; dual magnetron sputtering; thin-film deposition; photoelectron spectroscopy

## 1. Introduction

ZnWO$_4$ has been studied for the photocatalytic degradation of organic pollutants [1], for sacrificial water splitting [2] and as a scintillation material [3]. For these purposes, ZnWO$_4$ is most often fabricated as a powder using chemical synthesis procedures, such as hydrothermal [2,4], solvothermal [5], mechano–chemical [6] and sol-gel syntheses [7]. These chemical procedures have the advantage that the morphology of the ZnWO$_4$ particles can be tuned and that they can be produced in larger quantities.

Thin film-based materials prepared by thin-film deposition techniques allow, however, for a better integration in devices. It is relatively easily to deposit additional contact materials in a controlled and clean (vacuum) environment. ZnWO$_4$ thin films have already been successfully produced using spray pyrolysis [8] and dip coating [9].

There are, however, advantages of using magnetron sputtering compared to other thin-film deposition techniques, such as high sputter rates, easy control of process parameters and the ability to coat large areas [10]. Indeed, RF magnetron sputtering has already been used previously to prepare ZnWO$_4$ thin films by depositing a WO$_3$/ZnO/WO$_3$ heterolayer, followed by a post-annealing step [11]. However, ZnO crystallite impurities were found at each annealing temperature, indicating a poor stoichiometry control. In contrast to a single sputtering source, a dual magnetron sputtering setup has the additional advantage that the stoichiometry of ternary oxide films can be precisely controlled by altering the power on each target or by changing the target-to-substrate distance [12]. Other ternary oxides that have already been successfully deposited using dual magnetron sputtering include aluminum zirconate [13], bismuth niobate [14], zinc iridate [15], and zinc magnesium oxide [16].

In this work, a dual magnetron sputtering setup was used, using RF power and DC power to operate a ZnO target and a W target, respectively. Both pure $WO_3$ and ZnO have already been deposited using magnetron sputtering [17,18]. The deposition of a mixed Zn, W oxide in a dual magnetron sputtering procedure is, however, new. To create crystalline $ZnWO_4$ thin films, the ZnO sputtering power and oxygen content in the sputtering atmosphere were optimized to end up with stoichiometric $ZnWO_4$ thin films. After a post-calcination procedure, monoclinic $ZnWO_4$ could be obtained. The structural, morphological and chemical properties of the monoclinic $ZnWO_4$ thin films were characterized using X-ray photoelectron spectroscopy (XPS), scanning electron microscopy—energy dispersive X-ray spectroscopy (SEM-EDS), UV-Vis spectroscopy, X-ray diffraction (XRD) and Raman spectroscopy.

## 2. Materials and Methods

$ZnWO_4$ thin films were deposited using the dual magnetron sputtering setup at the DArmstadt Integrated SYstem for FUNdamental research (DAISY-FUN), described in greater detail elsewhere [19]. The ZnO target (2.0″ diameter, 99.99% purity, Mateck, Jülich, Germany) that operated with an RF signal at 13.56 MHz and the W target (2.0″ diameter, 99.95% purity, Lesker, Jefferson Hills, PA, USA) that operated with a DC signal were mounted onto 2.0″ MEIVAC MAK sputter deposition sources. To obtain stoichiometric thin films, the RF power given to the ZnO target was optimized to 55 W to match the sputter rate of the W target that operated at 25 W. The base pressure prior to deposition was in the $10^{-8}$ mbar range, and the standard operating atmosphere consisted of 50%/50% $Ar/O_2$ with a gas pressure of 0.005 mbar, which was maintained using a pressure feedback control. The thin films were grown on conductive (ca. 10 ohm/sq) $F:SnO_2$ (FTO) coated glass substrates and quartz glass substrates from Alineason (Frankfurt am Main, Germany). A sputter height of 20 cm was maintained, and substrates were rotated to ensure the homogeneity of the films. The thin films were deposited without intentional heating. Throughout the manuscript, the above process parameters are denoted as the standard parameters. Some of the samples also underwent a post-annealing step in a tube furnace operated under ambient atmosphere for 600 min at varying temperatures.

Photoelectron spectroscopy was used for surface analysis and was carried out with a SPECS Phoibos 150 analyzer (SPECS, Berlin, Germany), a SPECS Focus 500 X-ray source using the monochromatized $AlK_\alpha$ line at 1486.74 eV for X-ray photoelectron spectroscopy (XPS), and a He discharge lamp using the monochromatized HeII line at 40.8 eV for ultraviolet photoelectron spectroscopy (UPS). In order to determine the work function, a bias-voltage of $-3$ V was applied. The thin films were deposited on $F:SnO_2$/glass substrates to ensure a grounded electrical connection. Binding energies were calibrated by adjusting to the Ag $3d_{5/2}$-core level ($E_B$ = 368.3 eV), Cu $2p_{3/2}$-core level ($E_B$ = 932.7 eV) and Pt $4f_{7/2}$-core level ($E_B$ = 71.2 eV) from freshly sputter-cleaned samples.

Film thicknesses were measured using a Dektak XT profilometer (Bruker, Billerica, MA, USA). Thickness profiles were recorded by measuring with the profilometer tip from an uncoated area to a coated area on the substrate. The crystallinity of the $ZnWO_4$ thin films was determined through θ–2θ measurements using a Smartlab X-ray diffractometer equipped with a rotating Cu anode (Cu $K_\alpha$, 1.54 Å) (Rigaku, Tokyo, Japan). Raman spectra were recorded with a LabRAM HR Raman spectrometer (Horiba, Kyoto, Japan) using an Argon ion laser with a wavelength of 514.5 nm as the excitation source, yielding a laser power on the sample of approximately 10 mW, and a 10× objective, generating a spot size of approximately 2 μm. An XL30 FEG (Philips, Eindhoven, The Netherlands) was used for scanning electron microscopy (SEM) and energy-dispersive X-ray spectroscopy (EDS). UV-Vis optical spectroscopy was carried out using a Cary 7000 Universal Measurement Spectrophotometer (UMS) (Agilent, Santa Clara, CA, USA).

## 3. Results

The sputter parameters of WO$_3$ were kept fixed, while those of ZnO were allowed to change to match the sputter rate of WO$_3$. The sputter rate was calculated from the sputter time and the film thickness, which was deduced from profilometer measurements. As an example, the thickness profiles for WO$_3$ sputtered under standard conditions are shown in Figure S1. From the four profiles, a sputter rate of 2.3 nm/min can be deduced, considering a mean thickness of 105 nm and a deposition time of 45 min. To match this sputter rate, the RF power applied to the ZnO target was varied. Generally, sputter rates in RF sputtering are lower compared to DC sputtering [20]. The results of the varying sputter power on the ZnO film thickness can be seen in Figure 1a.

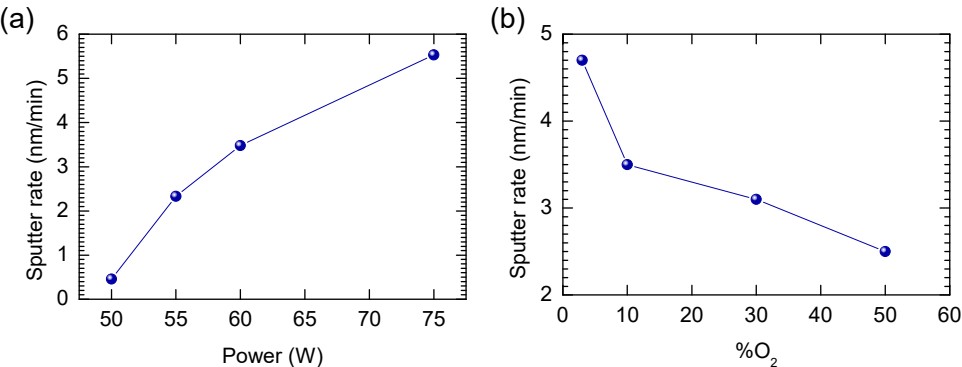

**Figure 1.** Sputter rate in dependence of (**a**) power change keeping other parameters fixed (50%/50% O$_2$/Ar) and (**b**) oxygen content change (rest Ar) keeping other parameters fixed (*p* = 60 W).

As expected, the sputter rate goes up when the ZnO sputtering power increases. At a ZnO RF sputtering power of 55 W and with 50% of O$_2$ in the process gas, a sputter rate of 2.3 nm/min was achieved, which matches the sputter rate of WO$_3$. Because now both WO$_3$ and ZnO can be sputtered at the same rate, stoichiometric ZnWO$_4$ should be obtained if both materials are sputtered at the same time. How the morphology of the ZnO thin films changes with respect to the ZnO sputtering power is shown in the SEM images in Figure S2. No noticeable morphological differences can be seen between the films deposited using different ZnO sputtering powers. Additionally, the oxygen content in the sputter gas was varied, keeping the ZnO RF sputtering power fixed at 60 W (Figure 1b). The sputter rate goes down as the oxygen content increases from 3% to 50%. Indeed, having less argon in the process atmosphere, which acts as the main sputter gas, generally leads to lower sputter rates [21,22]. However, high oxygen content is generally needed to avoid oxygen substoichiometry [23]. Therefore, by operating simultaneously the ZnO target at 55 W and the W target at 25 W in a 50%/50% argon/oxygen gas atmosphere stoichiometric thin films should be obtained. In Figure S3, the effect of the oxygen content on the morphology of the film is shown. In all samples, a grainlike structure can be observed. How the process gas atmosphere influences the ZnO crystallinity is shown in Figure S4. The limited differences between the X-ray diffractograms demonstrate, however, that the oxygen content during thin-film deposition has no significant influence on the ZnO crystallinity. One point of notice, however, is that arcing at the surface of the ZnO target could be observed after applying 50 Watt or more. Arcing is generally not preferred during sputtering as it can degrade the quality of the films [24]. However, the quality can be improved later during annealing. Figure 2 shows the effect of post-annealing at 600 °C for 10 h on the crystallinity of the ZnWO$_4$ thin films.

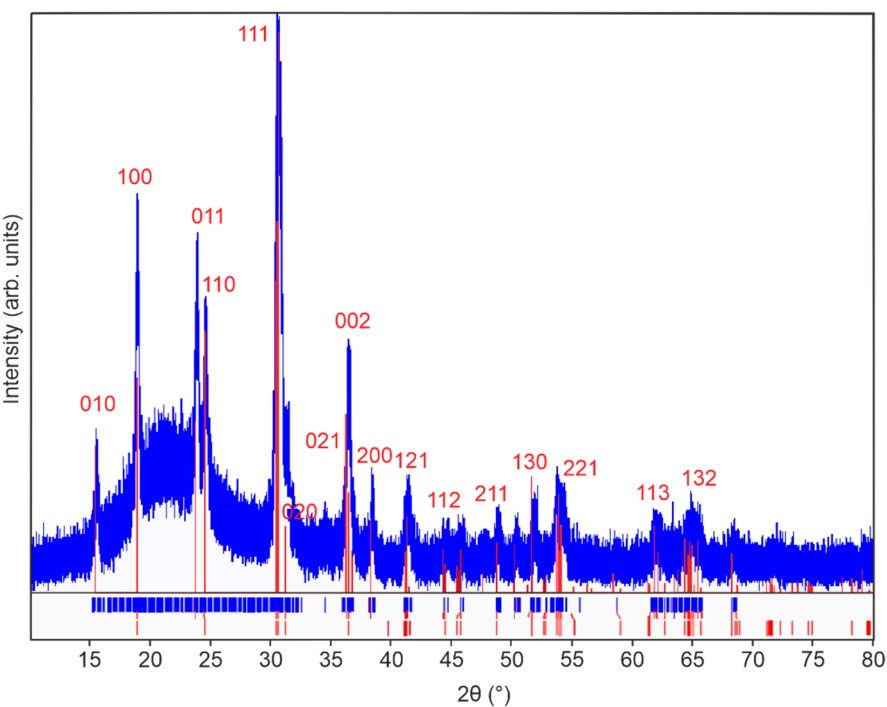

**Figure 2.** X-ray diffractogram of 100 nm monoclinic ZnWO$_4$ thin films annealed in air at 600 °C for 10 h. PDF card 96-591-0159 is used as reference and is shown in red.

The diffractogram of the sample annealed at 600 °C in air corresponds well to the monoclinic ZnWO$_4$ reference diffractogram (PDF card 96-591-0159). Since no additional reflexes can be observed, a phase pure monoclinic ZnWO$_4$ thin film can be assumed after annealing. Lower annealing temperatures did not result in phase pure ZnWO$_4$ (Figure S5). Raman spectroscopy was carried out to confirm the observations in the XRD analysis. Indeed, the Raman spectrum shown in Figure 3 appears to be characteristic for phase pure monoclinic ZnWO$_4$ with observable characteristic resonances at 910, 790, 712, 677, 549, 517, 410, 345, 316, 277, 198, 167, 150 and 126 cm$^{-1}$ [25]. The features at 910, 712, 549, 410, 345, 277, 198, and 126 cm$^{-1}$ can be attributed to A$_g$ vibrational modes, while the peaks at 790, 677, 517, 316, 167 and 150 cm$^{-1}$ correspond to B$_g$ vibrational modes [26]. Additional Raman spectra show that ZnWO$_4$ thin films annealed at lower temperatures are either amorphous (150 °C, 300 °C) or impure (450 °C) (Figure S6).

The X-ray photoelectron spectra for monoclinic ZnWO$_4$ can be found in Figure 4. Most of the peaks in the survey spectrum correspond to ZnWO$_4$. Because the spectra were recorded immediately after the post-annealing step, no carbon contamination can be noticed, as is evident from the absence of the C 1s signal at 285 eV. Na 1s could also be detected and likely originates from sodium in the borosilicate glass of the FTO/glass substrates that diffused out of the glass during the annealing step. The O 1s, Zn 2p$_{3/2}$ and W 4f$_{7/2}$ binding energies at 531.2 eV, 1022.4 eV and 36.3 eV are about 1 eV higher compared to other literature resources [9,27]. This 1 eV shift can be due to Na acting as a dopant leading to an increase of the Fermi level or due to surface charging, because the conductivity of ZnWO$_4$ is rather poor [28]. However, the core level lines do not exhibit any broadening as would be expected from surface charging [29]. There is a slight asymmetry in the O 1s profile towards higher binding energies, which could be related to the presence of surface-bound hydroxyl species. For a thinner amorphous ZnWO$_4$ film of about 14 nm, the binding energies were about 0.6 eV lower (Figure S7), which is much closer to earlier reported binding energy values. From the areas of the Zn 2p$_{3/2}$, W 4f and O 1s peaks in Figure S7, relative surface stoichiometries of 17%, 17% and 66% could be determined, which resembles stoichiometric ZnWO$_4$ thin films considering the typical 10% accuracy error in XPS quantification [30]. However, after annealing the surface stoichiometry changes to

31%, 7%, 56%, 6% for Zn, W, O, Na, respectively, which is a remarkable phenomenon since no ZnO phases could be detected during XRD or RAMAN measurements. Future studies should investigate this effect further. The surface stoichiometries are also summarized in Table 1.

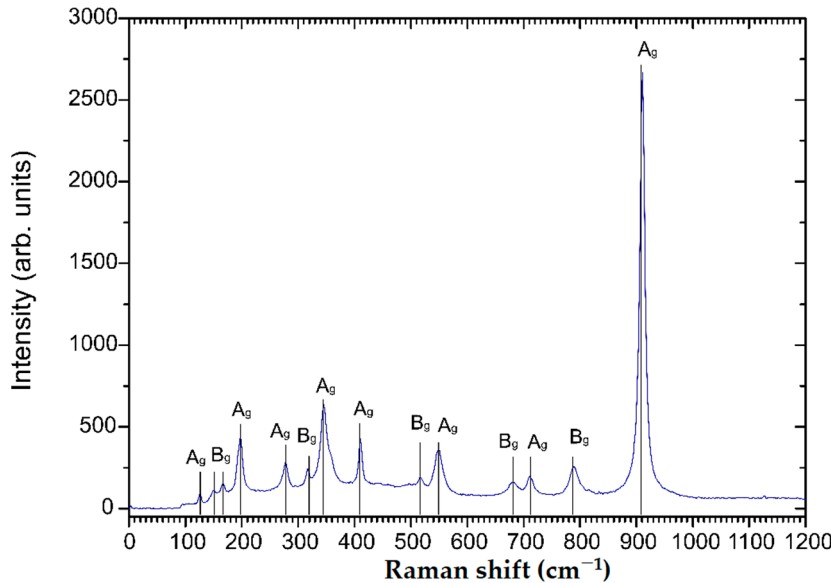

**Figure 3.** Raman spectrum of 100 nm monoclinic $ZnWO_4$ thin films annealed in air at 600 °C for 10 h.

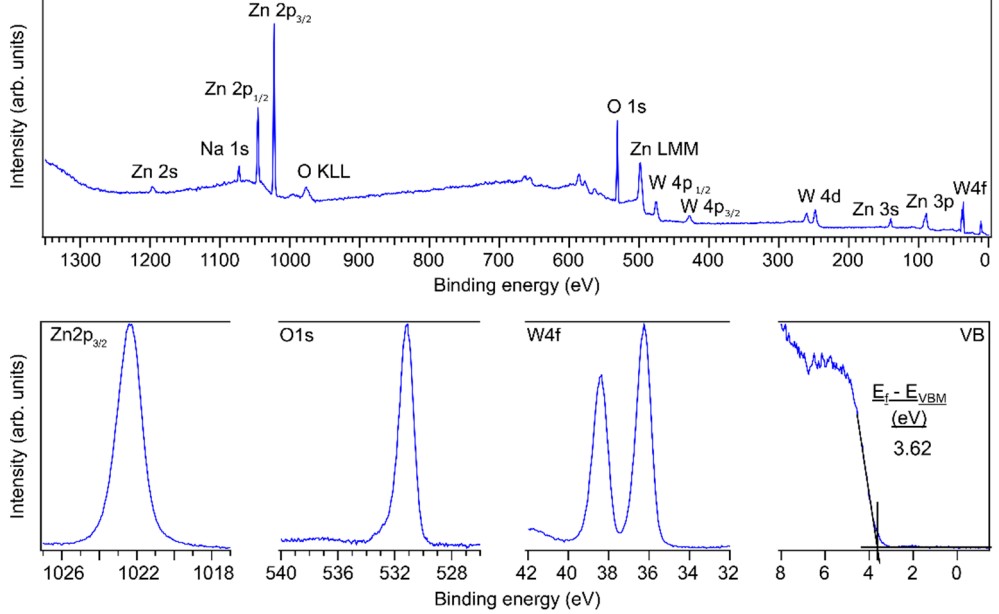

**Figure 4.** Survey, Zn $2p_{3/2}$, O 1s and W 4f core level, and valence band (VB) XP spectra of a 100 nm thick monoclinic $ZnWO_4$ film. Using tabulated sensitivity factors 2.77, 15.13 and 10.35 for O 1s, Zn $2p_{3/2}$ and W 4f the composition is 0.60, 0.21, 0.19.

**Table 1.** Surface stoichiometry amorphous and monoclinic $ZnWO_4$.

| $ZnWO_4$ | Zn $2p_{3/2}$ | W 4f | O 1s | Na 1s |
|---|---|---|---|---|
| amorphous | 17% | 17% | 66% | - |
| monoclinic | 31% | 7% | 56% | 6% |

To analyze the morphology and the bulk stoichiometry of the monoclinic $ZnWO_4$ thin films SEM-EDS was performed. In Figure S8, the morphology of the monoclinic $ZnWO_4$ film can be seen. The grains seem to be quite irregular and have sizes ranging from 10–100 nm. In Figure S9, the results of SEM-EDS spectroscopy are presented. As can be noticed, the quality of the SEM image is lower compared to the SEM image in Figure S8. The lower image quality is because a higher energy electron beam is used in SEM-EDS spectroscopy, which reduces the secondary electron imaging resolution also due to the rather poor conductivity of $ZnWO_4$. From the Zn K- and W L-edge EDS measurements, zinc and tungsten seem to be homogeneously distributed all over the sample. The bulk atomic percentages calculated from EDS, 39% and 61% for tungsten and zinc, respectively, are a bit off-stoichiometric. However, it is known that the accuracy of standard-less EDS spectroscopy after a ZAF matrix correction (as performed here) can be rather poor, amounting to a relative error up to 10% [31,32]. It would be best to compare the EDS measurements to that of a standard. For instance, $ZnWO_4$ single crystals with known stoichiometry could be used as standards. Although, the synthesis of $ZnWO_4$ single crystals has been reported before [3], they are not widely commercially available.

The optical band gap of monoclinic $ZnWO_4$ has been determined by UV−Vis optical spectroscopy (Figure S10). A Tauc plot of $(\alpha h\nu)^2$ vs. $h\nu$ yielded a direct optical band gap of 3.3 eV, which is similar to previous reports in the literature [33,34]. Nevertheless, strong variations of the $ZnWO_4$ optical band gap, depending on particular process parameters, have been published before [35]. It should provide a concise and precise description of the experimental results, their interpretation, and the experimental conclusions that can be drawn.

In addition, we have determined ultraviolet photoelectron (UP) spectra of the clean films (Figure 5a). The results show a Fermi level to VBM distance of 3.8 eV and 3.0 eV for monoclinic $ZnWO_4$ and amorphous $ZnWO_4$, respectively. The Fermi level to VBM distance determined for monoclinic $ZnWO_4$ is 0.2 eV higher compared to XPS, which could still be due to weak surface charging. From the secondary electron cutoffs, work functions amounting to 4.0 and 5.8 could be determined for monoclinic $ZnWO_4$ and amorphous $ZnWO_4$, respectively. Of interest with respect to the (opt)-electronic properties are the band energy diagrams that can be constructed from the UPS, XPS and optical data of the samples. These are shown in Figure 5b. The presented data indicate a highly n+ doped monoclinic $ZnWO_4$ with the Fermi level inside the conduction band. We attribute this result to the Na incorporation during the annealing process, which is estimated to be in the 3% range without leading to the formation of observable reduction of the $W^{6+}$ oxidation state. We also show in Figure 5 the values for the ionization potential and the electron affinity as they can be deduced from the UPS and optical absorption spectra. The apparent 1 eV difference in ionization potential and electron affinity between the amorphous and crystalline film is most likely due to the presence of a surface dipole. In contrast, the amorphous films show a Fermi-level position that is below the conduction band as found for many wide-band gap oxides, as is also schematically indicated in Figure 5b and as is deduced from the XPS and UPS data.

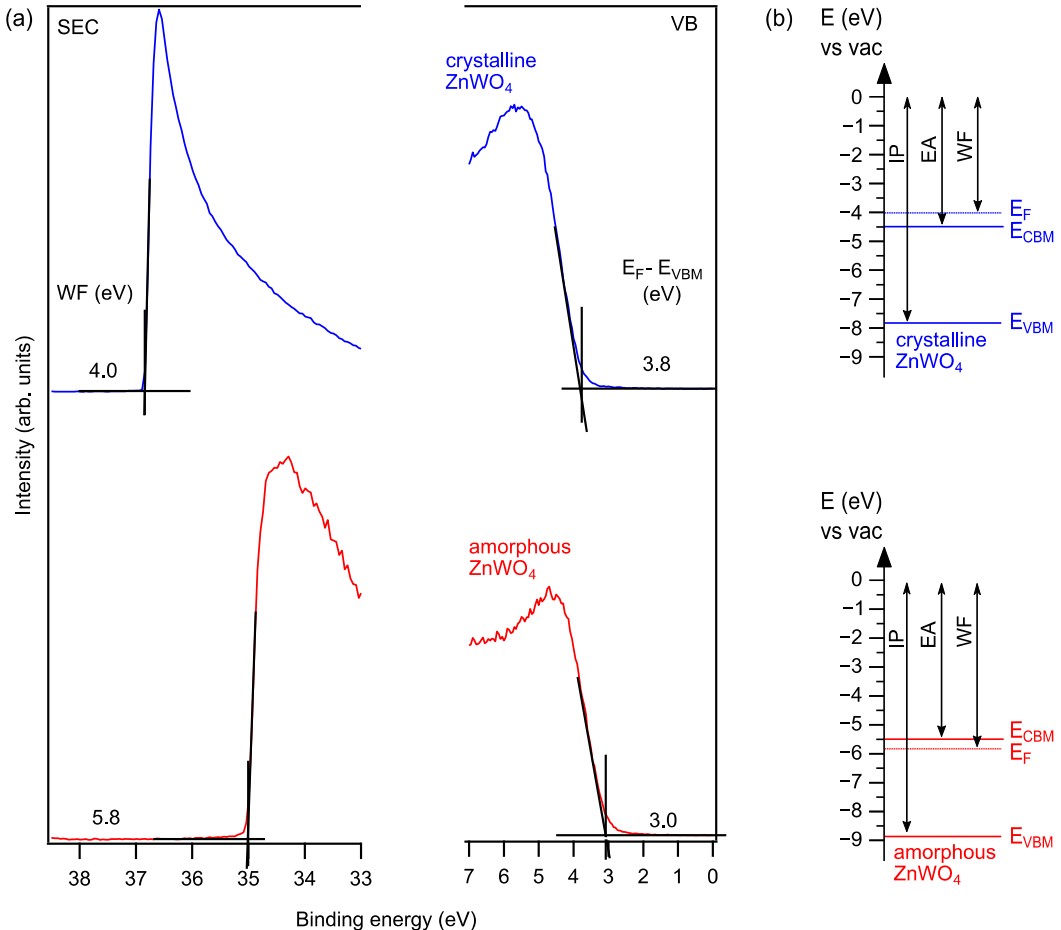

**Figure 5.** (**a**) UP spectra of the secondary electron cutoff (SEC) and valence band (VB) regions and (**b**) band diagrams of crystalline monoclinic $ZnWO_4$ and amorphous $ZnWO_4$. WF: work function; $E_F$: Fermi level; CBM: conduction band minimum; VBM: valence band maximum; IP: ionization potential; EA: electron affinity.

## 4. Conclusions

In this work, we have shown that monoclinic $ZnWO_4$ thin films can be prepared using a dual magnetron sputtering setup. The sputter rate of 2.3 nm/min of a W target operated using a DC signal at 25 W was matched by a ZnO target operated using an RF signal at 55 W. A subsequent post-annealing step holding the $ZnWO_4$ thin film for 10 h at 600 °C in air yielded monoclinic $ZnWO_4$. Future research efforts could focus on the elimination of arcing on the ZnO target. Due to the annealing step, Na is incorporated from the Na containing glass substrates leading to an n+ doping of the $ZnWO_4$. Future research efforts could focus on the magnetron sputtering of clean and structurally well-defined $ZnWO_4$ samples without extrinsic doping.

**Supplementary Materials:** The following are available online at https://www.mdpi.com/article/10.3390/surfaces4020013/s1: Sputter profiles of $WO_3$ thin films, SEM images and X-ray diffractograms of ZnO thin films, and SEM-EDS images, XP spectra and Tauc plot of $ZnWO_4$ thin films as well as additional information on the surface composition analysis using XPS; Figure S1: Sputter profiles of $WO_3$ deposited using standard DC sputter deposition parameters; Figure S2: SEM images of ZnO thin films deposited at different sputtering powers keeping other deposition parameters constant; Figure S3: SEM images of ZnO thin films deposited using different oxygen contents in the process gas keeping other deposition parameters constant; Figure S4 X-ray diffractograms of ZnO thin films deposited using different oxygen contents in the process gas keeping other deposition parameters constant; Figure S5: X-ray diffractograms of $ZnWO_4$ thin films annealed at different temperatures; Figure S6: Raman spectra of $ZnWO_4$ thin films annealed at different temperatures for 10 h in air;

Figure S7: Survey, Zn $2p_{3/2}$, O 1s and W 4f core level, and valence band (VB) XP spectra of a 15 nm thick amorphous $ZnWO_4$ film; Figure S8: SEM image of 100 nm monoclinic $ZnWO_4$ thin films annealed in air at 600 °C for 10 h; Figure S9: SEM-EDS of 100 nm monoclinic $ZnWO_4$ thin films annealed in air at 600 °C for 10 h; Figure S10: Tauc plot for the direct optical transition of the monoclinic $ZnWO_4$ thin film.

**Author Contributions:** Thin films were prepared and analyzed by Y.H. and F.M.; SEM-EDS measurements were performed by K.L.-W. and supervised by Y.H., T.M., and W.J.; Discussion and interpretation of results was conducted by Y.H., F.M., J.P.H., T.M., and W.J.; the original draft was composed by Y.H., edited by J.P.H., and revised by all authors; funding was acquired by T.M. and W.J. All authors have read and agreed to the published version of the manuscript.

**Funding:** Financial support through the project "Fundamentals of electro-chemical phase boundaries at semiconductor/electrolyte interfaces" GEP-HE funded by the German Federal Ministry of Education and Research BMBF under contract 13XP5023A is acknowledged.

**Institutional Review Board Statement:** Not applicable.

**Informed Consent Statement:** Not applicable.

**Data Availability Statement:** The data presented in this study are available free of charge from the corresponding author.

**Conflicts of Interest:** The authors declare no conflict of interest. The founding sponsors had no role in the design of the study; in the collection, analyses, or interpretation of data; in the writing of the manuscript; and in the decision to publish the results.

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
