# Peer review of "Reactive Dual Magnetron Sputtering: A Fast Method for Preparing Stoichiometric Microcrystalline ZnWO4 Thin Films"

_surfaces, doi:10.3390/surfaces4020013_

Round 1

Reviewer 1 Report

In this manuscript the authors reported a multi-technique study of ZnWo4 thin films, produced by a new approach using reactive dual magnetron sputtering. The manuscript is well structured and contains all the relevant information and references. The discussion and conclusions are supported by the data however I have some question concerning the innovative character of the approach and the stoichiometry.

At row 34 the authors claim that “magnetron sputtering has not yet been used for the fabrication of ZnWO$ thin films”: F.Zhang et al in “Structure and photoluminescence properties of ZnWO4 film prepared by depositing WO3/ZnO/WO3 heterolayer via magnetron sputtering technique” (Opt.Matl, vol85, p186, 2018 ) have shown the formation of ZnO4 by magnetron sputtering, already in 2018. An annealing procedure was used to obtain the desired material, as well as in this manuscript.

In my opinion the author should correct the statement at row 34, discussing possible improvements with respect to the approach of Zhang et al.

Moreover, from the Survey spectrum in Figure 4, the W4f signal appears quite low in intensity, even applying the mentioned sensitivity factors (15.13 and 10.35 for Zn2p 3/2 and W4f) it is hard to understand how the W4f core level signal becomes comparable to the Zn2p 3/2, giving that the stoichiometry should be 1:1. To the naked eye, the survey spectrum indicates a low W stoichiometry in the films. Can the authors explain this?

Two minor suggestion:

-labels in Figure 2 are quite small and difficult to read, i suggest to increase the font size.

-the sentence starting at row 30 is quite complex to read, could by simplified.

Author Response

Please find the comments in the attached word file

Reviewer 2 Report

Manuscript Number: Surfaces-1180264

Manuscript Title:” Reactive dual magnetron sputtering: a fast method for preparing stoichiometric microcrystalline ZnWO4 thin films”.

Recommendation: Accept after minor revision

Additional comments: The target of the manuscript is preparation of the effective ZnWO4 thin film photocatalyst and a scintillation material by dual magnetron sputtering.  This target is important and actual. So, the manuscript is dedicated to an actual problem of physical chemistry.

However, there are some problems in the manuscript that can be corrected.

  1. Supplementary materials are not reachable: “The webpage you are looking could not be found. The URL may have been incorrectly typed, or the page may have been moved into another part of the mdpi.com site.”
  2. It also will be useful to supply XRD pattern for the as-prepared sample of ZnWO4 thin film in Figure 2 along with annealed sample.
  3. The results of XPS, EDX analysis for the as-prepared and annealed samples can be shown in the Table in order to see the difference in oxygen and impurity content.

Author Response

You may find our comments in the attached word document

Reviewer 3 Report

Manuscript needs further improvement by incorporating the following suggestions:

1. It is not fully explained why the mode: 55 W for ZnO and 25 W for W targets with a working gas Ar:O (1:1) is optimal. What the authors mean: “the desired composition of ZnWO4 thin film” on line â„–117.

2. The work does not display micrographs and surface morphology of the films. The results obtained by the method of scanning electron microscopy are ubiquitous in similar works [1,2].

3. How only one annealing mode was chosen: 600 ℃ at 10 hours?

4. Why the results of physicochemical studies given only for one ZnWO4 film with a thickness of 100 nm and annealed at 600 ℃? Why the effect of annealing on investigated properties was not shown.

References

  1. Lou, Zhidong, Jianhua Hao, and Michael Cocivera. "Luminescence of ZnWO4 and CdWO4 thin films prepared by spray pyrolysis." Journal of luminescence 99.4 (2002): 349-354.
  2. Lou, Zhidong, and Michael Cocivera. "Cathodoluminescence of CaWO4 and SrWO4 thin films prepared by spray pyrolysis." Materials research bulletin 37.9 (2002): 1573-1582.

Author Response

Dear reviewer please find the answers to your review in the attached word file. Thank you again for your time in reviewing our manuscript

Round 2

Reviewer 3 Report

The authors' response provided significant data and information regarding the study.